# OUT-OF-DISTRIBUTION DETECTION WITH SMOOTH TRAINING

## ABSTRACT

Detecting out-of-distribution (OOD) inputs is important for ensuring the safe deployment of machine learning models in real-world scenarios. The primary factor impacting OOD detection is the neural network's overconfidence, where a trained neural network tends to make overly confident predictions for OOD samples. A naive solution to mitigate overconfidence problem is label smoothing. However, our experimental observations show that simply using label smoothing doesn't work. We believe that this is because label smoothing is applied to the original ID samples, which is the opposite of the goal of OOD detection (high confidence for ID samples and low confidence for OOD samples). To this end, we propose a new training strategy: smooth training (SMOT) where label smoothing is applied to the perturbed inputs. During the smooth training process, input images are masked with random-sized label-related regions, and their labels are softened to varying degrees depending on the size of masked regions. With this training approach, we make the prediction confidence of the neural network closely related to the number of input image features belonging to a known class, thus allowing the neural network to produce highly distinguishable confidence scores between in- and out-of-distribution data. Extensive experiments are conducted on diverse OOD detection benchmarks, showing the effectiveness of SMOT.

## 1 INTRODUCTION

Deep neural networks have achieved remarkable success thanks to the availability of large-scale labeled data. However, it's worth noting that most deep learning methods are designed within closed-set environments, where models are trained under the in-distribution (ID) assumption that the label space remains consistent during testing (Huang et al., 2017). In reality, situations may arise where samples from new classes spontaneously emerge, thus violating this assumption. To tackle this challenge, out-of-distribution (OOD) detection, proposed by (Bendale & Boult, 2016), is gaining increasing attention. In OOD detection, the model is not only expected to accurately classify ID samples but also to effectively distinguish OOD samples.

A pioneer method for detecting OOD samples is maximum softmax probability (MSP) (Hendrycks & Gimpel, 2017), where the maximum softmax probability is used as an indicator of OOD detection. Samples with low maximum softmax probability will be considered as OOD samples. However, neural networks tend to yield excessively confident predictions on OOD samples (Nguyen et al., 2015; Hein et al., 2019), making them less discriminative from ID samples. To this end, many representative methods (Lee et al., 2018b; Liu et al., 2020) attempt to design new OOD scoring functions to alleviate this overconfidence problem. In this paper, we consider solving this problem by *modifying the training strategy*.

We first investigate how humans determine the category of an object. Typically, humans do not always possess complete confidence in their judgment. As depicted in Figure 1, when humans view a complete cat, they can confidently identify it as such. But when only the tail of the cat is observed, no one can guarantee it is a cat. These conservative decisions make humans lower their confidence when encountering unknown objects. However, this is not the case when we train a neural network. For visual classification tasks, neural networks are typically trained with a cross-entropy loss. As optimizing the cross-entropy loss is proven to excessively increase the magnitude of the logits, networks tend to be *insensitive* to the absence of critical image region. As depicted in Figure

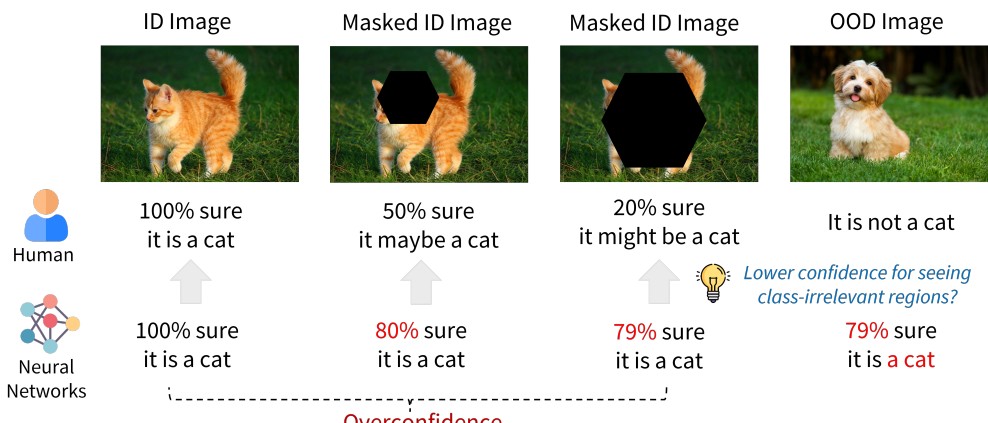

Figure 1: Humans don't always have complete confidence in their judgment, which allows them to make rational judgments when they encounter unknown objects.

2, an empirical study on ImageNet-1k (Deng et al., 2009) also evidence that: For a neural network pre-trained on ImageNet-1k, when we mask a portion of label-related regions of the images in the test set and feed them to the network, the neural network still makes high-confidence predictions for most of the masked images. These experimental observations show that neural networks trained with cross-entropy loss function are highly sensitive to the learned features. When some of the features belonging to a known class are observed (while not requiring all of them), the network makes high-confidence predictions. This may lead the network to make wrong detection results when the OOD samples have partially similar features to the ID samples.

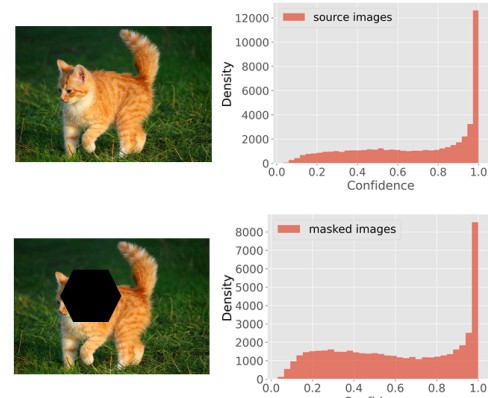

Figure 2: The network makes high-confidence predictions for the masked images.

In the realm of neural network training, a straightforward strategy for mitigating the overconfidence issue entails the application of label smoothing (Szegedy et al., 2016). This technique utilizes soft labels that are generated by harnessing a uniform distribution to smooth the distribution of the hard labels. However, adopting this approach may not necessarily enhance the model's capacity for detecting out-of-distribution instances and, in some cases, may even lead to a degradation in performance. This unfavorable outcome can be attributed to the fact that when the smoothing process is applied to the original ID samples, it systematically reduces their associated confidence scores. Consequently, these ID samples become less distinguishable from low-confidence OOD samples. Empirical studies conducted on the CIFAR-10 dataset have further substantiated the presence of this phenomenon as shown in Figure 3.

In this paper, we introduce a novel training strategy that improves out-of-distribution detection performance. In order to simulate the way humans perceive, we would like to feed the network images with different degrees of completeness during training, while softening theirs labels differently according to the degree of completeness. The challenge comes from getting images with different levels of completeness. Manually cropping each image in the training set is time-consuming, while randomly cropping or masking does not guarantee that the portion removed is label-relevant. To this end, we use Class Activation Maps (CAMs) (Zhou et al., 2016), a technique for visualizing neural networks that can obtain the contribution of different regions of an image to the predicted label. During training, we randomly mask different sizes of label-related regions on the input images and set different soft labels according to the size of the masked regions. We name our method Smooth Training (SMOT), distinct from the traditional training approach, in which the image labels are set to one-hot form throughout the training set (we call this training approach hard training). Intuitively, SMOT forces the neural network to give full confidence only when complete features belonging to an

ID class are detected, while the network adaptively outputs appropriate confidence when only some of the features are detected. This causes the network to output lower confidence when it observes an OOD sample with only partially the same features as ID classes, thus widening the confidence gap between ID and OOD samples. We summarize our contribution as follows:

- We show that one of the reasons for overconfidence in neural networks is the widely used cross-entropy loss function. Over-optimizing the cross-entropy loss function makes the network insensitive to missing features. When the OOD samples have partially similar features to the ID samples, the network gives overconfident predictions.

- To mitigate this problem, we propose a new training strategy: smooth training, by forcing the network to generate appropriate confidence based on the number of ID features detected, thus improving the ability of the model in handling OOD samples.

- Extensive experiments show that SMOT strategy greatly improves the OOD uncertainty estimation, and an ablation study is conducted to understand the efficacy of SMOT.

## 2 PRELIMINARIES

Let $\mathcal{X}$ and $\mathcal{Y} = \{1, \ldots, K\}$ represent the input space and ID label space, respectively. We consider the ID distribution $D_{X_{\mathrm{I}} Y_{\mathrm{I}}}$ as a joint distribution defined over $\mathcal{X} \times \mathcal{Y}$, where $X_{\mathrm{I}}$ and $Y_{\mathrm{I}}$ are random variables whose outputs are from spaces $\mathcal{X}$ and $\mathcal{Y}$. During testing, there are some OOD joint distributions $D_{X_{\mathrm{O}} Y_{\mathrm{O}}}$ defined over $\mathcal{X} \times \mathcal{Y}^c$, where $X_{\mathrm{O}}$ and $Y_{\mathrm{O}}$ are random variables whose outputs are from spaces $\mathcal{X}$ and $\mathcal{Y}^c$. Then following (Fang et al., 2022), OOD detection can be defined as follows:

**Problem 1** (OOD Detection). *Given sets of samples called the labeled ID data*

$$S = \{(\mathbf{x}^1, \mathbf{y}^1), \ldots, (\mathbf{x}^n, \mathbf{y}^n)\} \sim D_{X_{\mathrm{I}} Y_{\mathrm{I}}}^n, \ i.i.d.,$$

*the aim of OOD detection is to learn a predictor $g$ by using $S$ such that for any test data $\mathbf{x}$:*

- *if $\mathbf{x}$ is drawn from $D_{X_{\mathrm{I}}}$, then $g$ can classify $\mathbf{x}$ into correct ID classes;*

- *if $\mathbf{x}$ is drawn from $D_{X_{\mathrm{O}}}$, then $g$ can detect $\mathbf{x}$ as OOD data.*

Note that in problem 1, we use the one-hot vector to represent the label $\mathbf{y}$.

**Model and Risks.** In this work, we utilize $\mathbf{f}_{\boldsymbol{\theta}}$ to represent the deep model with parameters $\boldsymbol{\theta} \in \boldsymbol{\Theta}$, where $\boldsymbol{\Theta}$ denotes the parameter space. Let $\ell : \mathbb{R}^K \times \mathbb{R}^K \to \mathbb{R}_+$ be the loss function. Then, $R$ and $\widehat{R}$ are employed to denote the risk and empirical risk, respectively, i.e.,

$$R(\mathbf{f}_{\boldsymbol{\theta}}; D_{XY}) = \mathbb{E}_{(\mathbf{x}, \mathbf{y}) \sim D_{XY}} \ell(\mathbf{f}_{\boldsymbol{\theta}}(\mathbf{x}), \mathbf{y}), \quad \widehat{R}(\mathbf{f}_{\boldsymbol{\theta}}; S) = \frac{1}{n} \sum_{i=1}^{n} \ell(\mathbf{f}_{\boldsymbol{\theta}}(\mathbf{x}^i), \mathbf{y}^i).$$

**Score-based Strategy.** Many representative OOD detection methods (Hendrycks & Gimpel, 2017; Liang et al., 2018; Liu et al., 2020) follow a score-based strategy, i.e., given a model $\mathbf{f}_{\boldsymbol{\theta}}$ trained using $\mathcal{D}_{\mathrm{ID}}^{\mathrm{train}}$, a scoring function $S$ and a threshold $\tau$, then $\mathbf{x}$ is detected as ID data iff $S(\mathbf{x}; \mathbf{f}_{\boldsymbol{\theta}}) \geq \tau$:

$$G_\tau(\mathbf{x}) = \mathrm{ID}, \text{ if } S(\mathbf{x}; \mathbf{f}_{\boldsymbol{\theta}}) \geq \tau; \text{ otherwise, } G_\tau(\mathbf{x}) = \mathrm{OOD}. \tag{1}$$

In this paper, we use maximum softmax probability (MSP) (Hendrycks & Gimpel, 2017) as the scoring function to design our OOD detector, i.e.,

$$S_{\mathrm{MSP}}(\mathbf{x}; \mathbf{f}_{\boldsymbol{\theta}}) = \max_k \mathrm{softmax}_k(\mathbf{f}_{\boldsymbol{\theta}}), \tag{2}$$

where $\mathrm{softmax}_k(\mathbf{f}_{\boldsymbol{\theta}})$ is the $k$-th coordinate function of $\mathrm{softmax}(\mathbf{f}_{\boldsymbol{\theta}})$.

**Training Strategy.** In most score-based strategy, researchers mainly focus on designing effective scoring functions to extract the detection potential of deep model $\mathbf{f}_{\boldsymbol{\theta}}$. For the score-based methods, they follow a unified learning strategy—empirical risk minimization (ERM) principle, i.e.,

$$\min_{\boldsymbol{\theta} \in \boldsymbol{\Theta}} \widehat{R}(\mathbf{f}_{\boldsymbol{\theta}}; S). \tag{3}$$

In this work, our primary focus is to design a more effective training strategy that enhances the separation of ID and OOD data.

## 3 METHODOLOGY

In this section, we primarily present the main method Smooth Training.

### 3.1 OVERCONFIDENCE UNDER ERM

In this section, we explore the issue of overconfidence from a theoretical perspective using the ERM principle. Generally, the score-based OOD detection relies on a well-trained deep model, denoted as $\mathbf{f_\theta}$, which is trained based on the ERM principle (Eq. (3)), i.e.,

$$\boldsymbol{\theta}_S \in \arg\min_{\boldsymbol{\theta} \in \boldsymbol{\Theta}} \widehat{R}(\mathbf{f_\theta}; S).$$

**Overconfidence for ID Data.** Learning theory (Shalev-Shwartz & Ben-David, 2014) has indicated that when the model $\mathbf{f_\theta}$ has finite complexity, the risk of empirical predictor $\mathbf{f}_{\boldsymbol{\theta}_S}$ can approximate to the optimal risk with high probability, i.e.,

$$\mathbb{E}_{S \sim D^n_{X_I Y_I}} R(\mathbf{f}_{\boldsymbol{\theta}_S}; D_{X_I Y_I}) \leq \min_{\boldsymbol{\theta} \in \boldsymbol{\Theta}} R(\mathbf{f_\theta}; D_{X_I Y_I}) + \sqrt{\frac{C}{n}}, \tag{4}$$

where $C$ is a uniform constant. With this theoretical result, one can easily demonstrate that, under appropriate conditions, an overconfidence issue will arise for ID data. Theorem 1 provides a precise statement regarding the overconfidence issue in ID data.

**Theorem 1.** *Assume that the learning bound in Eq. (4) holds, and $\min_{\boldsymbol{\theta} \in \boldsymbol{\Theta}} R(\mathbf{f_\theta}; D_{X_I Y_I}) < \epsilon$, then for any $(\mathbf{x}, \mathbf{y}) \sim D_{X_I Y_I}$, with the probability at least $1 - \delta > 0$,*

$$\mathbb{E}_{S \sim D^n_{X_I Y_I}} \ell(\mathbf{f}_{\boldsymbol{\theta}_S}(\mathbf{x}), \mathbf{y}) \leq \frac{\epsilon}{\delta} + \sqrt{\frac{C}{\delta n}}.$$

Theorem 1 suggests that, given a sufficient amount of training data and a small optimal risk, i.e., $\min_{\boldsymbol{\theta} \in \boldsymbol{\Theta}} R(\mathbf{f_\theta}; D_{X_I Y_I})$, the issue of over-confidence for ID data is highly probable to arise.

**Overconfidence for OOD Data.** Building upon Theorem 1 and the insights from transfer learning as discussed in (Fang et al., 2021), we initiate a theoretical investigation into the overconfidence challenge associated with OOD data.

**Theorem 2.** *Let $d(\boldsymbol{\theta})$ be the disparity discrepancy between $D_{X_I}$ and $D_{X_O}$ (Fang et al., 2021), i.e., $d(\boldsymbol{\theta}) = \sup_{\mathbf{f}'} |\mathbb{E}_{\mathbf{x} \sim D_{X_I}} \ell(\mathbf{f_\theta}(\mathbf{x}), \mathbf{f}'(\mathbf{x})) - \mathbb{E}_{\mathbf{x} \sim D_{X_O}} \ell(\mathbf{f_\theta}(\mathbf{x}), \mathbf{f}'(\mathbf{x}))|$. If the conditions in Theorem 1 hold, then for any $(\mathbf{x}, \mathbf{y}) \in D_{X_O} \times D_{Y_I|X_I}$, with the probability at least $1 - \delta > 0$,*

$$\mathbb{E}_{S \sim D^n_{X_I Y_I}} \ell(\mathbf{f}_{\boldsymbol{\theta}_S}(\mathbf{x}), \mathbf{y}) \leq \frac{\epsilon + \mathbb{E}_{S \sim D^n_{X_I Y_I}} d(\boldsymbol{\theta}_S)}{\delta} + \sqrt{\frac{C}{\delta n}}.$$

Theorem 2 implies that the issue of overconfidence in OOD data under the ERM is primarily caused by three factors: 1) distribution discrepancy; 2) training data size; and 3) the optimal risk. When the distributions of ID and OOD have a smaller discrepancy, the issue of overconfidence becomes more severe. However, it is impossible to access real OOD data to reduce the distribution discrepancy during training. Additionally, when we only utilize limited training ID data, the issue of overfitting arises, leading to the failure of ID classification. Hence, this study primarily develops novel training strategy to 1) *achieve good performance on ID classification* (challenge 1), and 2) *mitigate the overconfidence issue induced by the small optimal risk* (challenge 2).

### 3.2 LABEL SMOOTH AND SMOOTH TRAINING

**Label Smooth.** Many methods attempt to address the issue of overconfidence by modifying the output of a well-trained neural network without the necessity of retraining (Liang et al., 2018; Sun et al., 2021; Zhu et al., 2022). Nevertheless, the effectiveness of these methods heavily relies on the selection of hyperparameters and can lead to prolonged detection times. In this work, we draw inspiration from a simple yet efficient method known as Label Smooth (Szegedy et al., 2016) to design our own approach. Label Smooth generates soft labels by utilizing a uniform distribution to smooth the distribution of the hard labels:

$$\mathbf{y}_\epsilon = (1 - \epsilon) \cdot \mathbf{y} + \epsilon/K \cdot \mathbf{u}, \tag{5}$$

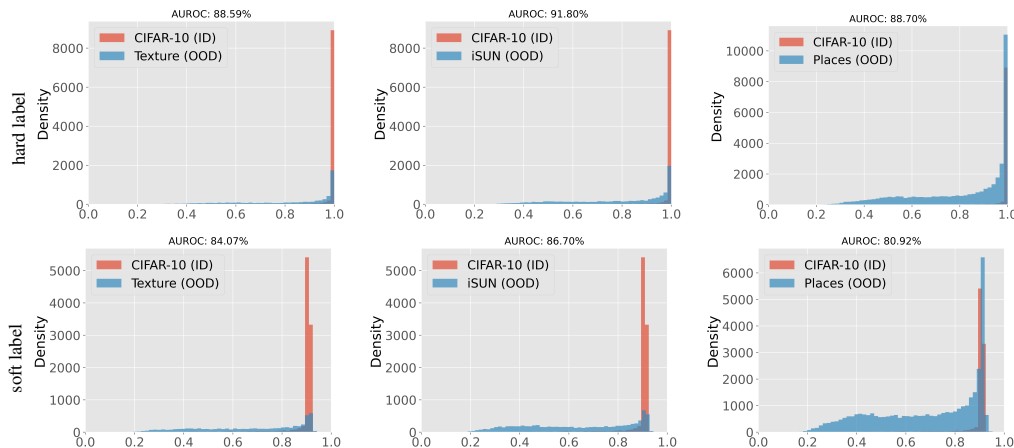

Figure 3: The MSP score distribution of training with hard target and training with soft target. Label smooth successfully mitigates the problem of overconfidence in neural networks, but does not lead to performance improvement on the OOD detection task.

where $\mathbf{u}$ is a uniform distribution across classes, and $\epsilon$ is denotes the smoothing parameter that is usually set to 0.1 in practice. However, both empirical results (see Figure 3) indicate that the sole use of Label Smooth cannot improve the performance of detecting OOD data.

**Smooth Training.** Applying Label Smooth directly to the original inputs results in low prediction confidence for ID data, leading a low ID classification performance. This goes against our initial purpose. Instead, we apply Label Smooth to the perturbed inputs, rather than the original inputs, and for the original inputs, we use hard labels (one-hot labels), i.e., given a perturbation function $T$,

$$\ell_T(\mathbf{x}, \mathbf{y}) = (1 - \lambda)\ell(f(\mathbf{x}), \mathbf{y}) + \lambda\ell(f(T(\mathbf{x})), \mathbf{y}_\epsilon), \tag{6}$$

where $\mathbf{y}_\epsilon = (1 - \epsilon) \cdot \mathbf{y} + \epsilon/K \cdot \mathbf{u}$ is the soft label. In Eq. (6), the term $\ell(f(\mathbf{x}), \mathbf{y})$ is used to address challenge 1 of achieving good ID classification. Additionally, the term $\ell(f(T(\mathbf{x})), \mathbf{y}_\epsilon)$ is used to address challenge 2 of mitigating the overconfidence issue induced by the small optimal risk.

The selection of the perturbation function is crucial for achieving smooth training. In the design of the perturbation function, it is important to adhere to the following two fundamental principles:

- *Effective.* Perturbations to the input must be able to cause perturbations in its true label.
- *Simple.* The perturbation function should be simple and have a low training cost.

**Perturbation by Masking.** In this study, we primarily employ the masking operation as the perturbation function. We anticipate that regions in an image with low correlation to the true label will exhibit lower confidence compared to regions with high correlation to the true label. As such, our primary focus is on masking those regions that have a strong correlation to the true label. To achieve this, we employ the CAM technique (Zhou et al., 2016), which is a weakly-supervised localization method that can identify discriminative regions.

Let $C_\mathbf{x}$ be the CAM of an input image $\mathbf{x}$. Then we obtain a mask by applying a masking threshold $t$ to the CAM, i.e.,

$$M(\mathbf{x}; t)[i, j] = \begin{cases} 0, & \text{if } C_\mathbf{x}[i, j] \geq t \\ 1, & \text{otherwise.} \end{cases} \tag{7}$$

By utilizing $T_m^t(\mathbf{x}) = M(\mathbf{x}; t) \cdot \mathbf{x}$ as the perturbation function, our loss function can be written as:

$$\ell_T(\mathbf{f_\theta}(\mathbf{x}), \mathbf{y}) = (1 - \lambda)\ell_{ce}(f(\mathbf{x}), \mathbf{y}) + \lambda\ell_{ce}(f(T_m^t(\mathbf{x})), \mathbf{y}_{\epsilon(t)}), \tag{8}$$

where $\ell_{ce}$ is the cross-entropy loss, and $\epsilon(t)$ is the smoothing parameter of Label Smooth and is designed as:

$$\epsilon(t) = 1 - \exp((t - 255)/T). \tag{9}$$

where $T$ is temperature coefficient. Then the final optimization problem is

$$\min_{\boldsymbol{\theta} \in \boldsymbol{\Theta}} \sum_{(\mathbf{x}, \mathbf{y}) \in S} \ell_T(\mathbf{f_\theta}(\mathbf{x}), \mathbf{y}). \tag{10}$$

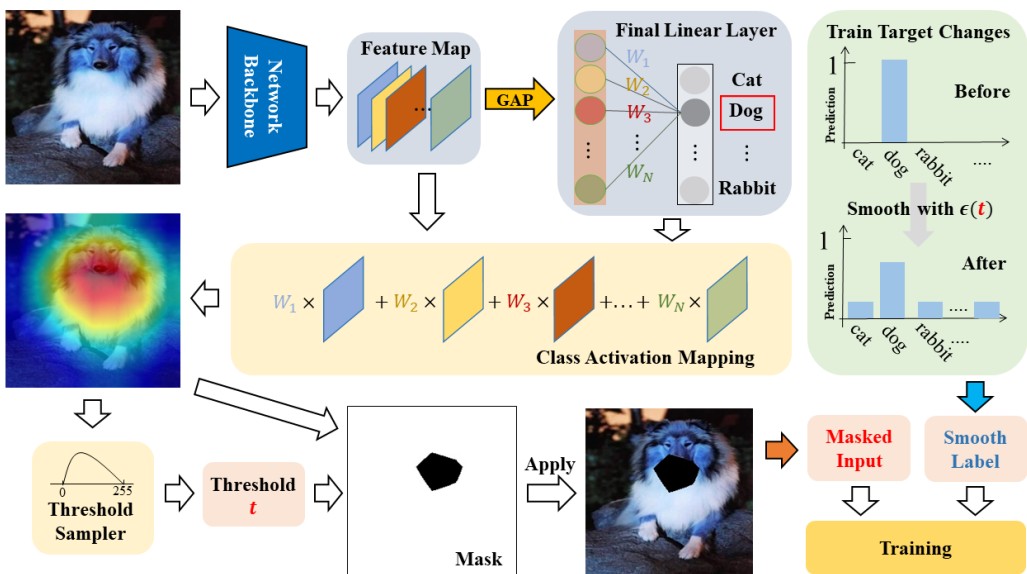

Figure 4: An overview of SMOT. We use a pre-trained network to obtain the heat map of the input image and then sample a threshold. The parts of the image with heat values greater than this threshold are masked and it's soft label is computed according to the threshold. The original image (with hard label) and the masked image (with soft label) are used simultaneously to train the final network.

In each iteration, we sample a masking threshold from a Beta distribution: $t \sim \text{Bata}(\alpha, \beta)$, and compute the risk in Eq. (10), using gradient backpropagation to update our model $f_\theta$. An overview of our method is presented in Figure 4. We refer to this training method as Smooth Training.

## 4 EXPERIMENTS

In this section, we present the performance comparison of the proposed method in the OOD detection scenario.

### 4.1 EXPERIMENT SETUP

**Datasets.** Following the common benchmark used in previous work (Zhang et al., 2023), we adopt CIFAR-10, CIFAR-100 (Krizhevsky et al., 2009) and ImageNet-200 (Deng et al., 2009) as our major ID datasets. For CIFAR10 and CIFAR100, all the images are of size $32 \times 32$. We use five common benchmarks as OOD test datasets: Textures (Cimpoi et al., 2014), SVHN (Netzer et al., 2011), iSUN (Xu et al., 2015), Places365 (Zhou et al., 2018) and LSUN (Yu et al., 2015). For all test datasets, the images are of size $32 \times 32$. And for ImageNet-200, all the image are of size $224 \times 224$. Following NPOS (Tao et al., 2023b), iNaturalist (Horn et al., 2018), SUN (Xiao et al., 2010), PLACES (Zhou et al., 2018), and TEXTURE (Cimpoi et al., 2014) are used as OOD test datasets.

**Evaluation metrics.** For evaluation, we follow the commonly-used metrics in OOD detection: (1) the false positive rate of OOD samples when the true positive rate of in-distribution samples is at 95%(FPR95), and (2) the area under the receiver operating characteristic curve (AUROC). We also report in-distribution classification accuracy (ID-ACC) to reflect the preservation level of the performance for the original classification task on ID data.

**OOD detection baselines.** We use both post-hoc inference methods and training methods as baselines. For post-hoc methods, we take MSP score (Hendrycks & Gimpel, 2017), ODIN (Liang et al., 2018), ReAct (Sun et al., 2021) and Energy score (Liu et al., 2020) as baselines. And for training methods, we use RegMixup (Pinto et al., 2022), VOS (Du et al., 2022), LogitNorm (Wei et al., 2022) and NPOS (Tao et al., 2023a) as baselines. Besides, we also compares the performance of SMOT under different OOD detection scoring functions.

Table 1: OOD detection performance comparison between using softmax cross-entropy loss and SMOT loss. All values are percentages. ↑ indicates larger values are better, and ↓ indicates smaller values are better. Bold numbers are superior results.

| ID datasets | CIFAR10 | | CIFAR100 | |
|---|---|---|---|---|
| OOD datasets | FPR95↓ | AUROC↑ | FPR95↓ | AUROC↑ |
| | Cross-entropy loss / **SMOT loss** | | Cross-entropy loss / **SMOT loss** | |
| Texture | 58.59/**23.01** | 88.59/**96.24** | 83.54/**73.83** | 77.89/**79.93** |
| SVHN | 55.71/**8.27** | 91.92/**98.21** | 60.61/**37.66** | 87.83/**93.52** |
| iSUN | 50.80/**12.27** | 91.80/**97.89** | 82.52/**64.10** | 73.52/**87.49** |
| Places | 57.85/**31.52** | 88.70/**93.88** | 81.12/**75.23** | 76.86/**78.62** |
| LSUN | 32.71/**2.04** | 95.33/**99.56** | 78.89/**68.88** | 81.72/**85.73** |
| Average | 51.13/**15.42** | 91.27/**97.15** | 77.34/**63.94** | 79.56/**85.06** |

Table 2: OOD detection performance on CIFAR10 as ID. Values are percentages. **Bold** numbers are superior results. ↑ indicates larger values are better, and ↓ indicates smaller values are better.

| | OOD Dataset | | | | | | | | | | | |
|---|---|---|---|---|---|---|---|---|---|---|---|---|
| | Texture | | SVHN | | iSUN | | Places | | LSUN | | Average | |
| Methods | FPR95↓ | AUROC↑ | FPR95↓ | AUROC↑ | FPR95↓ | AUROC↑ | FPR95↓ | AUROC↑ | FPR95↓ | AUROC↑ | FPR95↓ | AUROC↑ |
| MSP | 58.59 | 88.59 | 55.71 | 91.92 | 50.80 | 91.80 | 57.85 | 88.70 | 32.71 | 95.33 | 51.13 | 91.27 |
| ODIN | 51.96 | 88.82 | 48.33 | 92.41 | 41.42 | 92.56 | 49.93 | 89.29 | 21.2 | 96.66 | 42.56 | 91.95 |
| ReAct | 58.38 | 89.22 | 55.68 | 91.78 | 50.49 | 92.41 | 57.77 | 88.38 | 32.69 | 95.26 | 51.00 | 91.41 |
| Energy | 50.47 | 88.94 | 46.01 | 92.54 | 39.02 | 92.73 | 47.91 | 89.42 | 19.03 | 96.89 | 40.49 | 92.10 |
| RegMixup | 50.08 | 88.53 | 55.71 | 88.52 | 36.15 | 92.15 | 49.20 | 88.57 | 3.19 | 98.88 | 38.86 | 91.32 |
| VOS | 33.16 | 93.47 | 13.26 | 96.56 | 32.84 | 94.68 | 37.27 | 91.83 | 20.37 | 94.52 | 27.38 | 94.21 |
| LogitNorm | 31.98 | 94.28 | **2.94** | **99.31** | 13.24 | 97.67 | 34.88 | 93.56 | 2.45 | 99.43 | 17.10 | 96.85 |
| NPOS | 31.04 | 94.15 | 8.49 | 96.93 | 20.37 | 95.21 | 40.13 | 90.89 | 4.26 | 98.38 | 20.86 | 95.11 |
| SMOT | **23.01** | **96.24** | 8.27 | 98.21 | **12.27** | **97.89** | **31.52** | **93.88** | **2.04** | **99.56** | **15.42** | **97.15** |

**Training details.** For main results, we perform training with ResNet18 (He et al., 2016) on CIFAR-10, CIFAR100 and ImageNet200. For the base model, we train 200 epochs using SGD with cross-entropy loss, a momentum of 0.9, a weight decay of 0.0005, and a batch size of 128. We set the initial learning rate as 0.1 and divide it by 10 after 80 and 140 epochs. Then we train the final model with the proposed loss (Eq. (8)). We train 300 epochs using SGD, a momentum of 0.9, a weight decay of 0.0005, and a batch size of 128. The initial learning rate is 0.1, with cosine decay (Loshchilov & Hutter, 2017). The hyperparameters $\alpha$ and $\beta$ are set to 50 and 20. $\lambda$ is set to 0.1. For CIFAR10 CIFAR100 and ImageNet-200, we set T to 10, 150, and 200, respectively. Experiments are conducted on several NVIDIA GeForce RTX 2080, 3090 and 4090, using PyTorch.

## 4.2 RESULTS

**How does SMOT influence OOD detection performance?** In Table 1, we compare the OOD detection performance on models trained with cross-entropy loss and SMOT loss respectively. We keep the test-time OOD scoring function to be MSP. We observe that SMOT can effectively improve OOD detection performance. In CIFAR10 benchmark, SMOT reduces the average FPR95 from $51.13\%$ to $15.42\%$, a **35.71 %** of direct improvement. In CIFAR100 benchmark, SMOT also brings a improvement of 13.40% on average FPR95.

**Comparision with other baselines.** We conduct comparison experiments on CIFAR10, CIFAR100 and ImageNet-200 dataset. As shown in Table 2, 3, 4, on CIFAR10 and ImageNet-200, SMOT achieves the best average performance. On CIFAR100, SMOT also achieves good performance. The superior performance demonstrates the effectiveness of our training strategy. Smooth training successfully prevents overconfident predictions for OOD data, and improve test-time OOD detection.

## 4.3 ABLATION STUDY

**SMOT with different scoring functions**. In Table 5, we compare the OOD detection performance of neural networks trained with SMOT loss and cross-entropy loss under different scoring functions. Experimental results show that the OOD detection performance of the neural network trained with cross-entropy loss is influenced by the scoring function, while the OOD detection performance of the

Table 3: OOD detection performance on CIFAR100 as ID. Values are percentages. **Bold** numbers are superior results. ↑ indicates larger values are better, and ↓ indicates smaller values are better.

| Methods | Textures | | SVHN | | iSUN | | Places | | LSUN | | Average | |
|---|---|---|---|---|---|---|---|---|---|---|---|---|
| | FPR95↓ | AUROC↑ | FPR95↓ | AUROC↑ | FPR95↓ | AUROC↑ | FPR95↓ | AUROC↑ | FPR95↓ | AUROC↑ | FPR95↓ | AUROC↑ |
| MSP | 83.53 | 77.89 | 60.61 | 87.83 | 82.52 | 73.52 | 81.12 | 76.86 | 78.89 | 81.72 | 77.34 | 79.56 |
| ODIN | 83.26 | 77.76 | 56.21 | 89.89 | 80.10 | 78.37 | 81.66 | 77.02 | 79.97 | 83.32 | 76.24 | 81.27 |
| ReAct | 77.48 | 81.24 | 59.97 | 88.47 | 81.83 | 70.65 | 76.89 | **79.16** | 67.21 | 86.00 | 72.67 | 81.10 |
| Energy | 82.62 | 77.55 | 51.77 | 90.56 | 75.89 | 79.56 | 82.32 | 76.80 | 80.51 | 83.30 | 74.62 | 82.15 |
| RegMixup | 80.62 | 78.37 | 66.79 | 88.45 | 80.72 | 74.93 | 79.92 | 77.03 | 48.28 | 91.78 | 71.26 | 82.11 |
| VOS | 82.64 | 78.93 | 48.52 | 91.53 | 73.26 | 80.83 | 80.49 | 77.45 | 47.92 | 90.57 | 66.56 | 83.62 |
| LogitNorm | 80.05 | 76.19 | 47.26 | 92.43 | 95.18 | 67.88 | 81.27 | 76.68 | **11.00** | **98.05** | 62.95 | 82.24 |
| NPOS | **62.93** | **84.21** | **32.58** | 92.17 | 65.27 | 86.57 | **65.48** | 77.63 | 39.26 | 91.82 | **53.10** | **86.48** |
| SMOT | 73.83 | 79.93 | 37.66 | **93.52** | **64.10** | **87.49** | 75.23 | 78.62 | 68.88 | 85.73 | 63.94 | 85.06 |

Table 4: OOD detection performance on ImageNet-200 as ID. Values are percentages. **Bold** numbers are superior results. ↑ indicates larger values are better, and ↓ indicates smaller values are better.

| Methods | iNaturalist | | SUN | | Places | | Textures | | Average | |
|---|---|---|---|---|---|---|---|---|---|---|
| | FPR95↓ | AUROC↑ | FPR95↓ | AUROC↑ | FPR95↓ | AUROC↑ | FPR95↓ | AUROC↑ | FPR95↓ | AUROC↑ |
| MSP | 44.05 | 92.22 | 55.31 | 89.26 | 60.39 | 87.31 | 56.77 | 85.58 | 54.13 | 88.59 |
| ODIN | 41.41 | 93.13 | 52.48 | 91.04 | 58.42 | 88.74 | 49.11 | 88.33 | 50.36 | 90.31 |
| ReAct | 43.74 | 92.19 | 44.63 | 92.40 | 50.25 | 90.41 | 58.36 | 87.56 | 49.24 | 90.64 |
| Energy | 43.76 | 92.70 | 55.04 | 90.74 | 60.28 | 88.43 | 47.26 | 88.61 | 51.57 | 90.12 |
| RegMixup | 37.89 | 93.12 | 48.82 | 90.28 | 59.93 | 87.26 | 53.27 | 87.32 | 49.98 | 89.49 |
| VOS | 42.83 | 92.78 | 42.62 | 91.94 | 51.73 | 89.85 | 52.46 | 87.93 | 47.41 | 90.62 |
| LogitNorm | **17.63** | **96.18** | 43.92 | 91.27 | 47.82 | 90.28 | 35.83 | 89.73 | 36.29 | 91.86 |
| NPOS | 30.89 | 94.52 | 38.53 | 92.38 | 48.37 | 90.39 | **25.71** | **90.82** | 35.88 | 92.03 |
| SMOT | 24.83 | 95.82 | **27.72** | **93.17** | **45.83** | **90.92** | 34.29 | 89.74 | **33.04** | **92.35** |

neural network trained with SMOT loss is basically the same under different scoring functions. With SMOT, simply using MSP can achieve good OOD detection performance, thus saving testing time and not bothering with the choice of which scoring function to use.

**SMOT with different network architectures.** We test the performace of SMOT using ResNet-18 (He et al., 2016), WRN-40-2 (Zagoruyko & Komodakis, 2016) and DenseNet-BC (Huang et al., 2017). As shown in Table 6, SMOT can consistently improve OOD detection performance under different network structures, while being able to maintain the classification accuracy of ID samples.

**The effect of the sampling function.** In Table 7, we show how the parameters of the sampling function affects the OOD detection performance. The analysis is based on CIFAR-10. In general, a sampling distribution with higher variance leads to more diverse samples, and may require a larger network and longer training time to fit the training samples.

**The effect of temperature $T$.** In Figure 5, we further ablate how the parameter $T$ affects the OOD detection performance. The analysis is based on CIFAR-10. On this dataset, the best OOD detection performance is obtained when $T$ is set as 10. For CIFAR100 and ImageNet-200, we set a larger $T$. This is because when the number of classes is smaller, the neural network is more likely to be overconfident, in which case we should give a larger penalty.

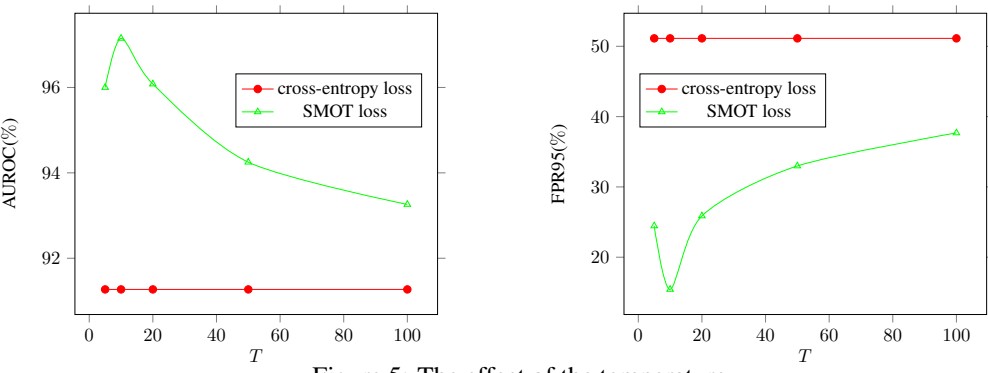

Figure 5: The effect of the temperature.

Table 5: SMOT with different scoring functions. Values are percentages.

| ID datasets | CIFAR10 | |
|---|---|---|
| Score | FPR95↓ | AUROC↑ |
| | Cross-entropy loss / SMOT loss | |
| Softmax | 51.13/15.42 | 91.27/97.15 |
| ODIN | 42.56/14.46 | 91.95/97.22 |
| Energy | 40.49/16.32 | 92.10/97.02 |
| ReAct | 51.00/15.42 | 91.41/97.15 |

Table 6: SMOT with different network architectures. Values are percentages.

| ID datasets | CIFAR10 | | |
|---|---|---|---|
| Architecture | FPR95↓ | AUROC↑ | ID ACC↑ |
| | Cross-entropy loss / SMOT loss | | |
| ResNet18 | 51.13/15.42 | 91.27/97.15 | 95.12/94.54 |
| WRN-40-2 | 49.50/27.38 | 91.30/94.98 | 95.08/94.62 |
| DenseNet | 51.15/29.53 | 89.28/94.76 | 94.61/93.79 |

Table 7: The effect of the sampling function.

| $Beat(\alpha, \beta)$ | Texture | | SVHN | | iSUN | | Places | | LSUN | | Average | |
|---|---|---|---|---|---|---|---|---|---|---|---|---|
| | FPR95↓ | AUROC↑ | FPR95↓ | AUROC↑ | FPR95↓ | AUROC↑ | FPR95↓ | AUROC↑ | FPR95↓ | AUROC↑ | FPR95↓ | AUROC↑ |
| w/o SMOT | 58.59 | 88.59 | 55.71 | 91.92 | 50.80 | 91.80 | 57.85 | 88.70 | 32.71 | 95.33 | 51.13 | 91.27 |
| (1,1) | 37.76 | 93.09 | 46.04 | 91.99 | 32.58 | 93.47 | 51.66 | 86.94 | 15.58 | 97.12 | 36.72 | 92.52 |
| (0.67,0.67) | 45.51 | 92.46 | 49.59 | 91.95 | 45.07 | 93.26 | 56.81 | 88.51 | 11.95 | 98.00 | 41.78 | 92.84 |
| (2,2) | 40.83 | 93.65 | 36.54 | 93.51 | 28.97 | 95.58 | 47.98 | 90.98 | 5.64 | 98.93 | 31.99 | 94.53 |
| (5,2) | 26.91 | 95.51 | 19.93 | 96.83 | 24.87 | 96.23 | 42.12 | 92.36 | 11.82 | 98.14 | 25.13 | 95.82 |
| (20,12) | 24.38 | 95.86 | 9.74 | 98.06 | 18.23 | 97.04 | 39.65 | 92.49 | 5.89 | 99.04 | 19.57 | 96.50 |
| (30,18) | 34.26 | 94.93 | 28.73 | 95.44 | 29.78 | 94.85 | 45.73 | 91.25 | 6.75 | 98.84 | 29.05 | 95.06 |
| (18,30) | 47.97 | 92.40 | 58.58 | 89.67 | 40.24 | 93.83 | 61.64 | 87.35 | 33.71 | 94.78 | 48.43 | 91.61 |
| (50,20) | 23.01 | 96.24 | 8.27 | 98.21 | 12.27 | 97.89 | 31.52 | 93.88 | 2.04 | 99.56 | 15.42 | 97.15 |
| (100,50) | 25.83 | 95.84 | 25.93 | 96.48 | 22.83 | 96.36 | 40.37 | 91.73 | 8.53 | 97.98 | 24.69 | 95.68 |

## 5 RELATED WORK

**Out-of-distribution detection.** The goal of OOD detection is to enable the model to distinguish between ID samples and OOD samples while maintaining the classification accuracy of ID samples. Many works try to mitigate the overconfidence of neural networks by designing different scoring functions, such as maximum softmax probability (Hendrycks & Gimpel, 2017), energy score (Liu et al., 2020), ReAct (Sun et al., 2021) and GradNorm score (Huang et al., 2021). Despite their simplicity and convenience, these methods are more like after-the-fact fixes. And, these types of methods may result in more detection time. In addition to that, the proposed scoring functions may have different effects in different scenarios, which sometimes need to be picked manually in practical applications. Some other approaches try to solve OOD detection problem by modifying the training strategy. For example, in (Lee et al., 2018a) (Hendrycks et al., 2019), the model is required to have uniform output over outliers. RegMixup (Pinto et al., 2022) utilizes Mixup (Zhang et al., 2018) as an additional regularizer to the standard cross-entropy loss. LogitNorm (Wei et al., 2022) enforces a constant vector norm on the logits in training.

**Confidence calibration.** Many previous works have shown that neural networks tend to be overconfident in their predictions (Hein et al., 2019) (Nguyen et al., 2015). To this end, some works address this problem by post-hoc methods such as Temperature Scaling (Platt et al., 1999). In addition, some method focus on the regularization of the model, such as weight decay (Guo et al., 2017), label smoothing (Szegedy et al., 2016). Our approach is an extension of label smoothing. By applying label smoothing to the perturbed inputs instead of the original inputs, the model is able to maintain a relatively high confidence in-distribution inputs.

## 6 CONCLUSION

In this paper, we propose Smooth Training (SMOT), a simple training strategy to enhance OOD detection performance. By modifying the labels of the training samples from single one-hot form to adaptive softened labels, the model tends to output conservative predictions, allowing the network to produce highly separable confidence scores for the ID and OOD samples. Extensive experiments show that SMOT can significantly improve OOD detection detection performance of the model while maintaining the classification accuracy of ID samples.

## ETHIC STATEMENT

This paper does not raise any ethical concerns. This study does not involve any human subjects, practices to data set releases, potentially harmful insights, methodologies and applications, potential conflicts of interest and sponsorship, discrimination/bias/fairness concerns, privacy and security issues, legal compliance, and research integrity issues.

## REPRODUCIBILITY STATEMENT

We summarize our efforts below to facilitate reproducible results:

- **Datasets.** We use publicly available datasets, which are described in detail in Section 4.1 and Appendix B.1.
- **Methodology.** Our method is fully documented in Section 3 and a complete theoretical proof is provided in Appendix A. Hyperparamters are specified in Section 4.1, with a thorough ablation study provided in Section 4.3.
- **Open Source.** Code will be available upon acceptance.

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

APPENDIX

# A PROOF

## A.1 PROOF OF THEOREM 1

*Proof.* Assume that $A_\delta \subset \mathcal{X} \times \mathcal{Y}$ is the area satisfying that for any $(\mathbf{x}, \mathbf{y}) \in A_\delta$

$$\mathbb{E}_{S \sim D^n_{X_I Y_I}} \ell(\mathbf{f}_{\boldsymbol{\theta}}(\mathbf{x}), \mathbf{y}) > \frac{\epsilon}{1 - \delta} + (\sqrt{\frac{C}{(1 - \delta)n}}.$$

Therefore,

$$D_{X_I Y_I}(A_\delta)\Big(\frac{\epsilon}{1 - \delta} + \sqrt{\frac{C}{(1 - \delta)n}}\Big) \le \epsilon + \sqrt{\frac{C}{n}},$$

which implies that $D_{X_I Y_I}(A_\delta) \le 1 - \delta$. We have completed this proof. $\qquad\square$

## A.2 PROOF OF THEOREM 2

*Proof.* Let $D'$ be the joint distribution whose marginal distribution is $D_{X_I}$ and conditional distribution is $D_{Y_I|X_I}$. Then

$$|\mathbb{E}_{S \sim D^n_{X_I Y_I}} R(\mathbf{f}_{\boldsymbol{\theta}_S}; D_{X_I Y_I}) - \mathbb{E}_{S \sim D^n_{X_I Y_I}} R(\mathbf{f}_{\boldsymbol{\theta}_S}; D')|$$
$$\le \mathbb{E}_{S \sim D^n_{X_I Y_I}} |R(\mathbf{f}_{\boldsymbol{\theta}_S}; D_{X_I Y_I}) - R(\mathbf{f}_{\boldsymbol{\theta}_S}; D')| \le \mathbb{E}_{S \sim D^n_{X_I Y_I}} d(\boldsymbol{\theta}_S).$$

Then using the same proving process of Theorem 1, we complete this proof. $\qquad\square$

# B DETAILS OF DATASETS

In this section, we provide dataset details.

## B.1 IMAGENET-200 BENCHMARK

Following OpenOOD (Zhang et al., 2023), we choose 200 classes from ImageNet-1k (Deng et al., 2009) to create ImageNet-200. The chosen classes are the same as OpenOOD:

n01443537, n01484850, n01494475, n01498041, n01514859, n01518878, n01531178, n01534433, n01614925, n01616318, n01630670, n01632777, n01644373, n01677366, n01694178, n01748264, n01770393, n01774750, n01784675, n01806143, n01820546, n01833805, n01843383, n01847000, n01855672, n01860187, n01882714, n01910747, n01944390, n01983481, n01986214, n02007558, n02009912, n02051845, n02056570, n02066245, n02071294, n02077923, n02085620, n02086240, n02088094, n02088238, n02088364, n02088466, n02091032, n02091134, n02092339, n02094433, n02096585, n02097298, n02098286, n02099601, n02099712, n02102318, n02106030, n02106166, n02106550, n02106662, n02108089, n02108915, n02109525, n02110185, n02110341, n02110958, n02112018, n02112137, n02113023, n02113624, n02113799, n02114367, n02117135, n02119022, n02123045, n02128385, n02128757, n02129165, n02129604, n02130308, n02134084, n02138441, n02165456, n02190166, n02206856, n02219486, n02226429, n02233338, n02236044, n02268443, n02279972, n02317335, n02325366, n02346627, n02356798, n02363005, n02364673, n02391049, n02395406, n02398521, n02410509, n02423022, n02437616, n02445715, n02447366, n02480495, n02480855, n02481823, n02483362, n02486410, n02510455, n02526121, n02607072, n02655020, n02672831, n02701002, n02749479, n02769748, n02793495, n02797295, n02802426, n02808440, n02814860, n02823750, n02841315, n02843684, n02883205, n02906734, n02909870, n02939185, n02948072, n02950826, n02951358, n02966193, n02980441, n02992529, n03124170, n03272010, n03345487, n03372029, n03424325, n03452741, n03467068, n03481172, n03494278, n03495258, n03498962, n03594945, n03602883, n03630383, n03649909, n03676483, n03710193, n03773504, n03775071, n03888257, n03930630, n03947888, n04086273, n04118538, n04133789, n04141076, n04146614, n04147183, n04192698, n04254680, n04266014, n04275548, n04310018, n04325704, n04347754, n04389033, n04409515, n04465501, n04487394, n04522168, n04536866, n04552348,

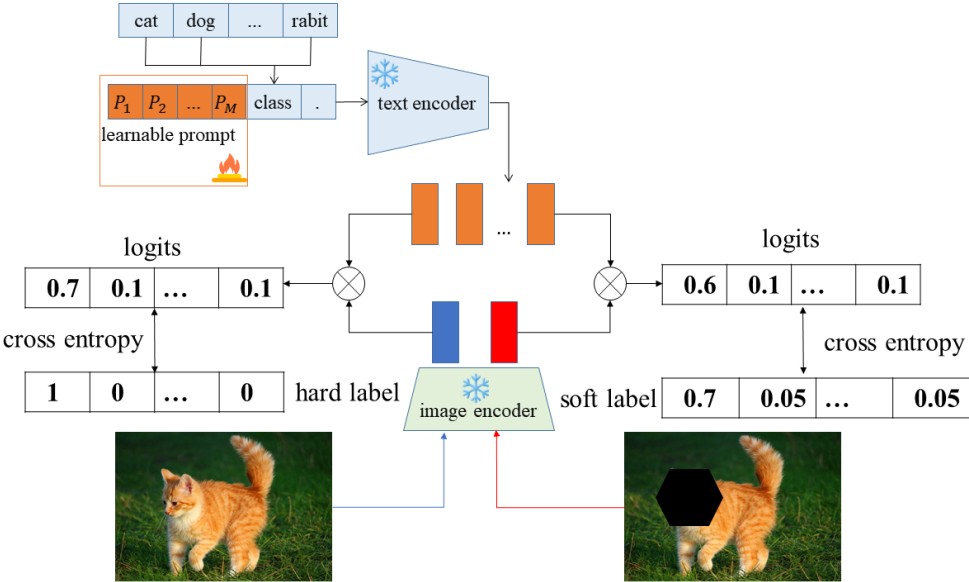

Figure 6: SMOT running on CLIP. The image encoder and text encoder are frozen. We use SMOT loss to learn the prompt.

n04591713, n07614500, n07693725, n07695742, n07697313, n07697537, n07714571, n07714990, n07718472, n07720875, n07734744, n07742313, n07745940, n07749582, n07753275, n07753592, n07768694, n07873807, n07880968, n07920052, n09472597, n09835506, n10565667, n12267677.

We use subsets from iNaturalist (Horn et al., 2018), SUN (Xiao et al., 2010), Places (Zhou et al., 2018) and Texture (Cimpoi et al., 2014) as OOD datasets, which are created by Huang *et al.* (Huang & Li, 2021). The classes from OOD datasets do not overlap with ImageNet-1k. A brief description about them is as follows:

**iNaturalist** contains images from natural world. It has 13 super-categories and 5089 sub-categories covering plants, insects, birds, mammals, and so on. The subset containing 110 plant classes not showing in ImageNet-1k are chosen as OOD test set.

**SUN** contains 899 categories that cover more than indoor, urban, and natural places. We use the subset which contains 50 natural objects not overlapping with ImageNet-1k.

**Places** contains photos labeled with scene semantic categories from three macro-classes: Indoor, Nature, and Urban. We use subset sampled from 50 categories that are not present in ImageNet-1k.

**Texture** contains images of textures and abstracted patterns. As no categories overlap with ImageNet-1k, we use the entire dataset.

## C    RESULTS ON IMAGENET-1K

Training a network with SMOT loss from scratch on ImageNet-1k is too expensive. Instead, we leverage CLIP (Radford et al., 2021). Our approach is the same as CoOp (Zhou et al., 2022), which is known as visual prompt learning. During the training process,clip's image encoder and text encoder are frozen and only a small number of ID samples are used to learn the input of the text encoder. The training process is shown in Figure 6. Each class, we use 16 samples to learn the prompt. We compare SMOT with the following baselines, MCM (Ming et al., 2022), MSP (Fort et al., 2021), ODIN  (Liang et al., 2018), Energy  (Liu et al., 2020), GradNorm  (Huang et al., 2021), Vim  (Wang et al., 2022), KNN  (Sun et al., 2022), VOS (Du et al., 2022), NPOS (Tao et al., 2023b) and CoOp (Zhou et al., 2022). For convenience, we use the pre-trained ResNet-18 to obtain the heat maps. As shown in Table 8, SMOT achieves the best performance with only a small number of training samples.

Table 8: OOD detection performance on ImageNet-1k as ID. Except for our method SMOT, all other experimental results are from NPOS (Tao et al., 2023b).

| Methods | iNaturalist | | SUN | | Places | | Textures | | Average | |
|---|---|---|---|---|---|---|---|---|---|---|
| | FPR95↓ | AUROC↑ | FPR95↓ | AUROC↑ | FPR95↓ | AUROC↑ | FPR95↓ | AUROC↑ | FPR95↓ | AUROC↑ |
| MCM | 32.08 | 94.41 | 39.21 | 92.28 | 44.88 | 89.83 | 58.05 | 85.96 | 43.55 | 90.62 |
| MSP | 54.05 | 87.43 | 73.37 | 78.03 | 72.98 | 78.03 | 68.85 | 79.06 | 67.31 | 80.64 |
| ODIN | 30.22 | 94.65 | 54.04 | 87.17 | 55.06 | 85.54 | 51.67 | 87.85 | 47.75 | 88.80 |
| Energy | 29.75 | 94.68 | 53.18 | 87.33 | 56.40 | 85.60 | 51.35 | 88.00 | 47.67 | 88.90 |
| GradNorm | 81.50 | 72.56 | 82.00 | 72.86 | 80.41 | 73.70 | 79.36 | 70.26 | 80.82 | 72.35 |
| Vim | 32.19 | 93.16 | 54.01 | 87.19 | 60.67 | 83.75 | 53.94 | 87.18 | 50.20 | 87.82 |
| KNN | 29.17 | 94.52 | 35.62 | 92.67 | 39.61 | 91.02 | 64.35 | 85.67 | 42.19 | 90.97 |
| VOS | 31.65 | 94.53 | 43.03 | 91.92 | 41.62 | 90.23 | 56.67 | 86.74 | 43.24 | 90.86 |
| VOS+ | 28.99 | 94.62 | 36.88 | 92.57 | 38.39 | 91.23 | 61.02 | 86.33 | 41.32 | 91.19 |
| NPOS | **16.58** | **96.19** | 43.77 | 90.44 | 45.27 | 89.44 | 46.12 | 88.80 | 37.93 | 91.22 |
| CoOp | 30.21 | 94.63 | 33.46 | 93.13 | 40.56 | 90.18 | 56.78 | 87.25 | 40.25 | 91.30 |
| SMOT | 20.45 | 95.83 | **31.27** | **93.73** | **35.72** | **91.83** | **42.47** | **89.37** | **32.48** | **92.69** |

## D  THE EFFECT OF TEMPERATURE $T$ ON CIFAR100.

In Figure 7, we further ablate how the parameter T affects the OOD detection performance on CIFAR100 dataset. On this dataset, the best OOD detection performance is obtained when T is set as 150.

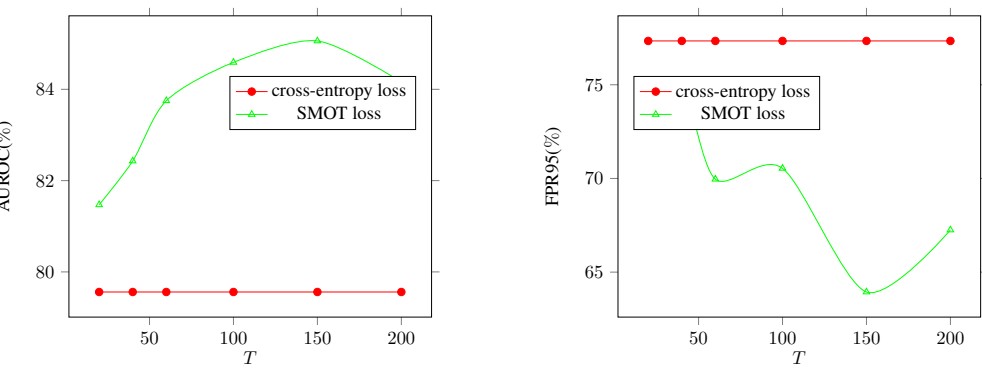

Figure 7: The effect of the temperature.

## E  SMOT WITH OTHER PERTURBATION FUNCTION.

In the formal paper, we use mask as the perturbation function. Here, We try to perturb the image by adding gaussian noise to it (while softening its label). In addition, we also add a set of experiments in which the masked part is set to 1. As shown in Table 9, Using different perturbation functions can all improve the OOD detection performance of the model.

Table 9: SMOT with different perturbation function.

| Methods | Texture | | SVHN | | iSUN | | Places | | LSUN | | Average | |
|---|---|---|---|---|---|---|---|---|---|---|---|---|
| | FPR95↓ | AUROC↑ | FPR95↓ | AUROC↑ | FPR95↓ | AUROC↑ | FPR95↓ | AUROC↑ | FPR95↓ | AUROC↑ | FPR95↓ | AUROC↑ |
| MSP | 58.59 | 88.59 | 55.71 | 91.92 | 50.80 | 91.80 | 57.85 | 88.70 | 32.71 | 95.33 | 51.13 | 91.27 |
| SMOT (masked as 0) | 23.01 | 96.24 | 8.27 | 98.21 | 12.27 | 97.89 | 31.52 | 93.88 | 2.04 | 99.56 | 15.42 | 97.15 |
| SMOT (masked as 1) | 37.73 | 93.63 | 34.82 | 94.11 | 26.98 | 95.89 | 42.28 | 92.31 | 15.34 | 97.67 | 31.43 | 94.72 |
| SMOT (noise) | 41.40 | 93.05 | 36.78 | 94.64 | 27.91 | 95.87 | 44.64 | 91.40 | 23.4 | 96.77 | 34.68 | 94.35 |

## F  COMPARISON OF THE DISTRIBUTION OF MSP SCORES.

We compare the MSP scores distribution on CIFAR10 benchmark for networks trained using cross-entropy loss, label smoothing, and SMOT loss, respectively. As shown in Figure 8, the three columns

from left to right are the results of cross-entropy loss, label smoothing, and SMOT loss, respectively. It can be observed that the networks trained with SMOT loss have low confidence for most of OOD samples. Compared to cross-entropy loss and label smoothing, SMOT produces more distinguishable confidence for ID and OOD samples.

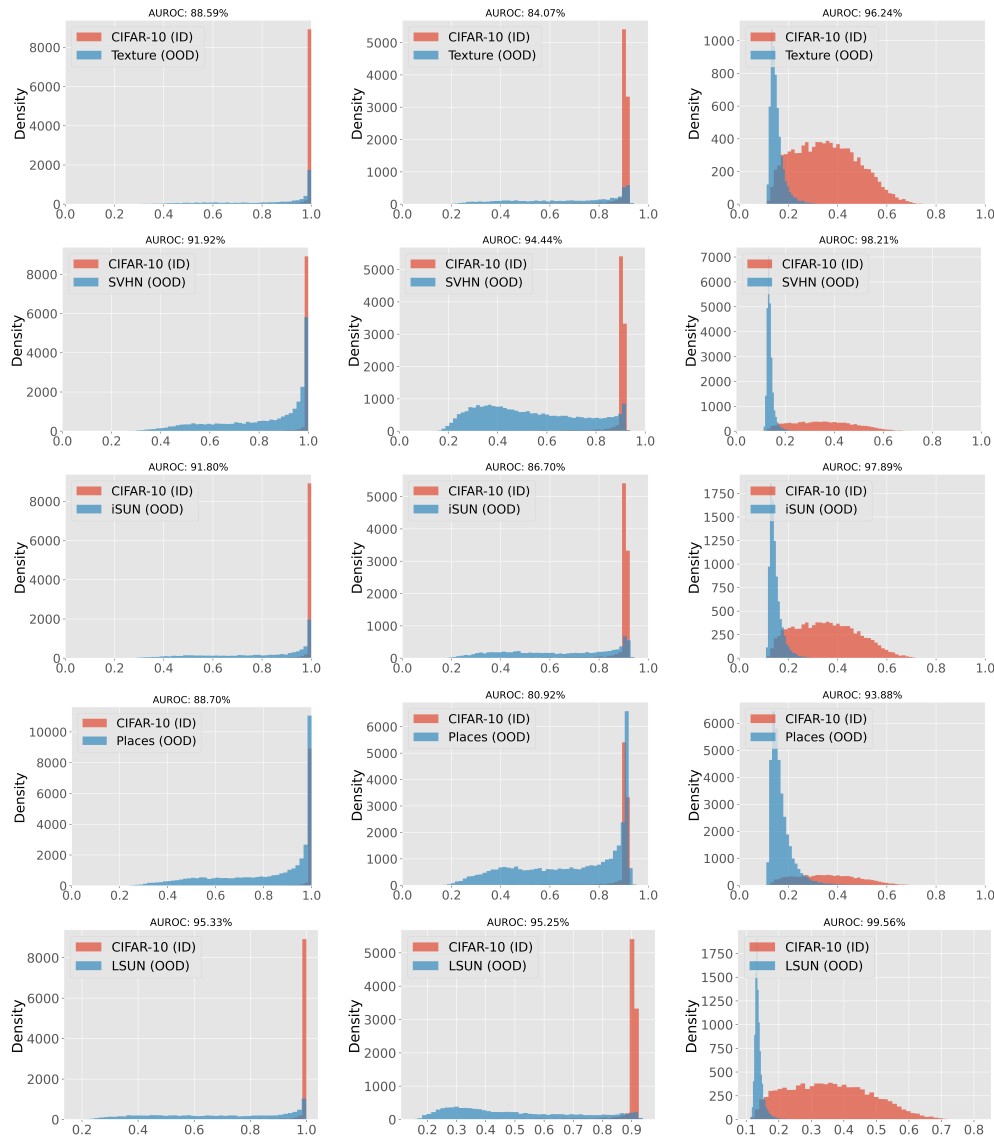

Figure 8: The MSP score distribution of training with cross-entropy loss, label smoothing and SMOT loss. From left to right are the results of cross-entropy loss, label smoothing, and SMOT loss.

## G  QUALITATIVE RESULTS.

Qualitatively, we show the t-SNE visualization of the features generated by by the networks trained with cross-entropy loss and SMOT loss respectively. As shown in Figure 10 and Figure 11, the network trained by SMOT loss reduces the number of OOD samples which are deep in clusters of ID classes.

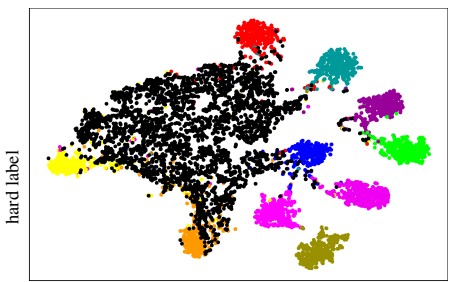 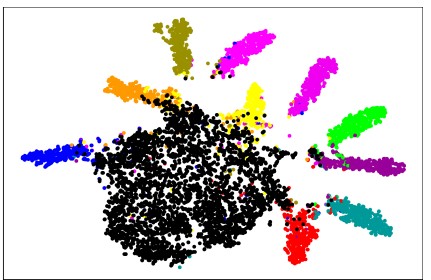

Figure 10: t-SNE visualization of features generated by network trained with cross-entropy loss.

Figure 11: t-SNE visualization of features generated by network trained with SMOT loss.

