# OpenReview forum: "Out-Of-Distribution Detection With Smooth Training"
_ICLR.cc/2024/Conference — Submitted to ICLR 2024_

### Official Review · Reviewer_c4yF · 2023-10-19

**Soundness:** 3 good
**Presentation:** 4 excellent
**Contribution:** 3 good
**Rating:** 6
**Confidence:** 3

**Summary:**

This paper addresses the issue of label smoothing in OOD detection. It first identifies the cause of overconfidence prediction is cross-entropy loss in neural networks. It then proposes a new training scheme of label smoothing, SMOT, for perturbed inputs. SMOT proposes to train models on the confidence of different areas of mask-out regions.

**Strengths:**

1. It applied label smoothing beyond training data where it demonstrated labeling smoothing on ID data is not enough.
2. The proposed training method without using auxiliary OOD dataset, but perturbation of masking is very promising where OOD auxiliary dataset is not available.
3. The supplementary sections justify many decisions made in the main paper, such as why masking is chosen. The paper is generally complete and clear.

**Weaknesses:**

The discussions/conclusions from Theorem 1 and 2 are too abrupt and not very obvious, such as the paragraph just before the Sec.3.2. More explanations are needed, especially for Theorem 2. It makes it less self-contained.

**Questions:**

1. What are the intuitions/interpretations of $\sqrt{\frac{C}{n}}$ in Equ(4), and the equations of equations of theorem 1 and 2? The connection between the equations and the implications is not clear.
2. Can SMOT use the updated model trained on the fly to get the mask for perturbation?

---

> ### Author Response · Authors · 2023-11-23
>
> >**Q1:** What are the intuitions/interpretations of $\sqrt(C/n)$ in Equ(4), and the equations of equations of theorem 1 and 2? The connection between the equations and the implications is not clear.
>
> **A1:** We apologize for the confusion. $n$ is the number of training samples and $C$ is a uniform constant. When the number of the training samples is large enough, this term tends to 0 such that the risk of empirical predictor $\mathbf{f}_{\boldsymbol{\theta}_S}$  can approximate to the optimal risk.
> Theorem 1 states that under the right conditions, the model is likely to be overconfident in the ID data. Theorem 2 states that when the model is overconfident in the ID data, it will also be overconfident in the OOD data which have a small distribution discrepancy with ID data. We'll go into more detail in the revised version.
>
> >**Q2:** Can SMOT use the updated model trained on the fly to get the mask for perturbation?
>
> **A2:** Thanks for your kind suggestion, we perform the experiment you described. We still train 300 epochs, in the first 100 epochs we perform the standard training and then we use the model in training to obtain the CAM to perform the SMOT training. The results are as follows:
> |                         | Avg FPR95 | Avg AUROC |
> |-------------------------|-----------|-----------|
> | msp                     | 51.13     | 91.27     |
> | SMOT trained on the fly | 25.05     | 95.44     |
>
> It is shown what you described works well! Thanks again for your insightful comments. We will add these results and discussions to our revision.

---

### Official Review · Reviewer_m2VV · 2023-10-28

**Soundness:** 2 fair
**Presentation:** 2 fair
**Contribution:** 2 fair
**Rating:** 5
**Confidence:** 5

**Summary:**

This paper proposes SMOT, a smooth training algorithm for OOD detection. SMOT is based on the heuristic that masking out certain features from the input image should correspondingly leads to decrease in the network's prediction confidence. Specifically, SMOT leverages CAM together with random-thresholding to determine the masking region, and the soft label (or essentially prediction confidence encoded in the training target) is determined according to the threshold (which is related to the area of the masking region if I understand correctly). Experiments on CIFAR-10/100 and ImageNet-200 show that SMOT exhibits (moderate) performance improvements over certain existing methods.

**Strengths:**

- The manuscript is in general clearly written.

**Weaknesses:**

### Theoretical Investigation

I find that Sec. 3.1 is somewhat hard to follow. The message / motivation it tries to convey is unclear to me. See specific comments or questions below.

1. The conclusion of Theorem 1 is "given a sufficient amount of training data and a small optimal risk, ..., the issue of over-confidence for ID data is highly probable to arise". However, the equation is only related to "over-confidence" (which I assume refers to excessive maximum softmax probability according to Eq. 2) when the loss is exactly cross-entropy loss. If we use label smoothing as the loss (although it is later empirically shown not to work), then there won't be over-confidence by looking at Eq. 4.

2. My same argument could be applied to Theorem 2 as well. Furthermore, I can't see how exactly the "over-confidence in OOD data" is reflected in Theorem 2. More elaboration and clarification is necessary.

3. The concluding paragraph under Theorem 2 makes me lost again. Why we want to "access real OOD data to reduce the distribution discrepancy during training"? What does it mean to "reduce the distribution discrepancy" (the $d(\theta)$ in Theorem 2?) between ID and OOD? Meanwhile, why suddenly "limited training ID data", "overfitting", and "the failure of ID classification" become issues for OOD detection?

4. Lastly, where are the proofs of the Theorems (or where are the references if they were proved by existing works)?



### Design of SMOT

1. Eq. 9 seems a little arbitrary. Why using a temperature-scaled exponential function? What's the intuition behind it? Why (t - 255)? What is the value range of t?

### Experiments

1. One limitation of the experiments and presented results is the fact that all considered OOD datasets are far-OOD which are easier to be detected. I expect to see more results on near-OOD splits (e.g., CIFAR-100 or Tiny ImageNet for CIFAR-10, SSB or NINCO for ImageNet), which are more likely to translate to real-world where the OOD images can be extremely similar to ID images.

2. The baseline selection seems a bit arbitrary. How does SMOT compare with recent top-performing methods (e.g., ASH [1] as identified by OpenOOD [2])? Also, a highly relevant baseline is missing (see below "Related Work" for details).

3. Why the training budget and learning rate scheduler is different between base models and SMOT models? Specifically, base models are trained for 200 epochs, while the "final model" with the proposed SMOT loss is trained for 300 epochs). Meanwhile, the base model adopts a step-wise learning rate decay schedule, while the final model uses the more advanced cosine decay. Is this a fair comparison, especially given that both longer training and sophisticated scheduler exactly benefit OOD detection (Table 5 in [3])?

4. Lastly, an important ablation study that I believe should be included is how SMOT compares with random masking / cropping. This would better justify SMOT's design of leveraging CAM to determine the masking region.

### Related Work
Sec. 5 should be more thorough and informative. Specifically, notice that the general idea of using corrupted / perturbed images associated with soft labels has been explored in at least two works in the field of OOD detection [4, 5]. Among these, [4] is in particular relevant to this work. I put up a table below making high-level comparison between [4] and this work.

|      | soft target |  perturbation  | needs a pre-trained model? |
|------|-------------|---|------|
| [4]  |  $y_\epsilon=(1-\epsilon)\cdot y + \epsilon / K \cdot u$ (see their Eq. 3)  |   image corruptions defined by ImageNet-C   | a pre-trained classifier for determining $\epsilon$ |
| SMOT |  $y_\epsilon=(1-\epsilon)\cdot y + \epsilon / K \cdot u$ (Eq. 5 in this work)   |  masking  | a pre-trained classifier for generating CAM mask |

From the above table, it is not obvious what advantages SMOT can offer over [4] (e.g., less compute, not requiring a pre-trained model). Therefore, I believe that [4] should be not only referenced but also included as an actual baseline to show that CAM-based masking and the associated method for assigning $\epsilon$ value are better than the designs of [4].

### Format
The references are inserted absurdly (e.g. "ResNet18 He et al. (2016)") which I believe are not in the most appropriate format. There are also some mis-formatting, e.g. "Eq.equation 8" in "Training details".

------------------------------------------------------
[1]  Extremely simple activation shaping for out-of-distribution detection

[2] OpenOOD v1.5: Enhanced Benchmark for Out-of-Distribution Detection

[3] Open-Set Recognition: A Good Closed-Set Classifier Is All You Need

[4] Bridging In- and Out-of-distribution Samples for Their Better Discriminability

[5] Mixture Outlier Exposure: Towards Out-of-Distribution Detection in Fine-grained Environments

**Questions:**

Please see the questions in Weaknesses.

---

> ### Author Response · Authors · 2023-11-23
>
> We sincerely thank you for your constructive comments! Please find our responses below.
>
> >**Q1:** The conclusion of Theorem 1 is "given a sufficient amount of training data and a small optimal risk, ..., the issue of over-confidence for ID data is highly probable to arise". However, the equation is only related to "over-confidence" (which I assume refers to excessive maximum softmax probability according to Eq. 2) when the loss is exactly cross-entropy loss. If we use label smoothing as the loss (although it is later empirically shown not to work), then there won't be over-confidence by looking at Eq. 4.
>
> **A1:** Thanks for your valuable comments.
> Theorem 1 states that networks trained on ERM principle are likely to be overconfident in ID data. We do not discuss the vanilla label smoothing here. We agree that label smoothing can alleviate the problem of overconfidence in neural networks. However, this is more of a conclusion drawn from experimental results. It is hard to follow the your conclusion "*If we use label smoothing as the loss, then there won't be over-confidence by looking at Eq. 4*". In fact, label smoothing is not widely used in OOD detection. Our experimental results also demonstrate that vanilla label smoothing does not improve the model's OOD detection performance. We intuitively believe that this may be due to the fact that label smoothing is applied to raw ID samples, which is contrary to the goal of OOD detection.
>
> >**Q2:** I can't see how exactly the "over-confidence in OOD data" is reflected in Theorem 2. More elaboration and clarification is necessary.
>
> **A2:** We apologize for the misunderstanding.
> Theorem 2 states that for any OOD sample $x$, the upper bound of the risk that the well-trained model
>  $\mathbf{f}_{\boldsymbol{\theta}_S}$ misassigns it to the label space of the ID samples is the right-hand term of the inequality in Theorem 2 (I'm sorry, but openreview doesn't seem to be able to compile that equation).
>
> >**Q3:** The concluding paragraph under Theorem 2 makes me lost again. Why we want to "access real OOD data to reduce the distribution discrepancy during training"? What does it mean to "reduce the distribution discrepancy" (the $d(\theta)$ in Theorem 2?) between ID and OOD? Meanwhile, why suddenly "limited training ID data", "overfitting", and "the failure of ID classification" become issues for OOD detection?
>
> **A3:** Thanks for your insightful comments.
>
> In the OE approach, we are allowed to use surrogate OOD data to regularize the model, i.e., to allow the model to have low confidence on this data. However, it is clear that the surrogate OOD data has distributional differences from the real OOD data encountered during testing. Here REDUCE the distribution discrepancy refers to reducing the distribution discrepancy between the surrogate OOD data and the real OOD data.
>
>
> >**Q4:** Lastly, where are the proofs of the Theorems (or where are the references if they were proved by existing works)?
>
> **A4:** The proofs are in APPENDIX A. We will highlight it in our revision. Thanks for your kind suggestion.
>
> >**Q5:** Eq. 9 seems a little arbitrary. Why using a temperature-scaled exponential function? What's the intuition behind it? Why (t - 255)? What is the value range of t?
>
> **A5:** Thanks for pointing out this potentially confusing problem. We design the smoothing parameter function based on the following three principles：
>
> - The smoothing parameter should be a monotonically decreasing function of the masking threshold, because the more regions are masked, the smoother its label should be.
> - When no region is masked, the smoothing parameter should be (or close to) 0.
> - When all regions are masked, the smoothing parameter should be (or close to) 1.
>
> The designed temperature-scaled exponential function satisfies the above rules and can simply adjust the steepness of the function by one parameter $T$. Of course, other more complex functions or learnable functions can be considered, and this is our future work. $t$ is the masking threshold, ranging from 0 to 255, corresponding to the values in the generated heat map. We set $(t-255)$ because when the masking threshold $t$ is 255, the image will not be masked, and we should use hard label, i.e., $\epsilon = 0$.

---

> > ### Author Response · Authors · 2023-11-23
> >
> > >**Q6:** One limitation of the experiments and presented results is the fact that all considered OOD datasets are far-OOD which are easier to be detected. I expect to see more results on near-OOD splits (e.g., CIFAR-100 or Tiny ImageNet for CIFAR-10, SSB or NINCO for ImageNet), which are more likely to translate to real-world where the OOD images can be extremely similar to ID images.
> >
> > **A6:** Thank you for your suggestion. We supplement experiments under the near-OOD setting, where we compared the OOD detection ability of model trained using SMOT loss with that of a model trained with normal cross-entropy loss, using CIFAR10 as the ID dataset and CIFAR100 and tinyimagenet as the OOD dataset, respectively, with the following results:
> > |               | CIFAR100 |       | Tiny-ImageNet |       |
> > |---------------|----------|-------|---------------|-------|
> > |               | FPR95    | AUROC | FPR95         | AUROC |
> > | cross entropy | 62.03    | 87.31 | 59.68         | 87.23 |
> > | SMOT          | 48.14    | 90.69 | 42.29         | 91.30 |
> >
> > Apparently SMOT works well under  near-OOD setting as well.
> >
> > >**Q7:** The baseline selection seems a bit arbitrary. How does SMOT compare with recent top-performing methods (e.g., ASH [1] as identified by OpenOOD [2])? Also, a highly relevant baseline is missing (see below "Related Work" for details).
> >
> > **A7:** Following your constructive comments, we compared ASH and SMOT on CIFAR10, and the results show that SMOT can outperform ASH:
> >
> > |      | Avg FPR95 | Avg AUROC |
> > |------|-----------|-----------|
> > | ASH  | 52.17     | 91.04     |
> > | SMOT | 15.42     | 97.15     |
> >
> > For the other two works you mentioned, they do inspire us a lot, especially [4]. The use of the classification accuracy of corrupted data on the base model to assign labels to the corrupted data is novel to us. But since the authors don't seem to have published it in conferences or journals, it has very few citations and we haven't noticed this work before. The biggest difference between ours and [4] is that we use masking as perturbation function, and we believe that our approach is more in line with the way humans perceive the world. Also our label assignment function is continuous while theirs is discrete. We are not sure which of these two approaches is better. We will explore this work more closely in a revised version later. We tentatively compared the performance of SMOT with [4] and [5] on CIFAR10, as follows:
> > |          | Avg FPR95 | Avg AUROC |
> > |----------|-----------|-----------|
> > | [4]      | 18.27     | 96.84     |
> > | MixOE[5] | 13.55     | 97.59     |
> > | SMOT     | 15.42     | 97.15     |
> >
> > As you can see, our method is slightly stronger than [4], but none of them can beat MixOE, which is reasonable because MixOE uses an extra dataset. Thanks again for your careful review!
> >
> >
> > >**Q8:** Why the training budget and learning rate scheduler is different between base models and SMOT models? Specifically, base models are trained for 200 epochs, while the "final model" with the proposed SMOT loss is trained for 300 epochs). Meanwhile, the base model adopts a step-wise learning rate decay schedule, while the final model uses the more advanced cosine decay. Is this a fair comparison, especially given that both longer training and sophisticated scheduler exactly benefit OOD detection (Table 5 in [3])?
> >
> > **A8:** Thanks for the heads up. We train our base model following the standard way of training a CIFAR10 classification model. When training the final model, we intuitively traine more epochs due to the increased diversity of samples and choose cosine decay due to the fact that it does not require tuning. We re-trained the base model with the same way we train the final model and find that it does lead to a small performance improvement(average AUROC from 91.27% to 92.02 and average FPR95 from 51.13% to 48.53%). We will correct the corresponding baselines in our revision for a fairer comparison.
> >
> > >**Q9:** Lastly, an important ablation study that I believe should be included is how SMOT compares with random masking / cropping. This would better justify SMOT's design of leveraging CAM to determine the masking region.
> >
> > **A9:** Thanks to your suggestion, we have added the experiment, with the randomly generated masks. Please see **A3** to **Reviewer H2kX**.
> >
> > >**Q10:** The references are inserted absurdly (e.g. "ResNet18 He et al. (2016)") which I believe are not in the most appropriate format. There are also some mis-formatting, e.g. "Eq.equation 8" in "Training details".
> >
> > **A10:** Thanks for the heads up, we've corrected the error in the revised version.

---

### Official Review · Reviewer_2XDU · 2023-10-31

**Soundness:** 3 good
**Presentation:** 3 good
**Contribution:** 3 good
**Rating:** 6
**Confidence:** 4

**Summary:**

This paper proposes label smoothing training framework for OOD detection. The authors use the CAM to identify the regions that have a strong correlation to the true label, and generate a masked input image and corresponding soft label for smooth training. Extensive experiments show that the smooth training strategy greatly improves the OOD performance with different score functions.

**Strengths:**

1. The proposed smooth training (SMOT) strategy, where soft labels are applied to the perturbed inputs, is technical sound to relieve overconfidence problem.
2. The image masking and label smoothing strategy is quite novel and makes sense.
3. The paper is well structured and in good presentation and writing.

**Weaknesses:**

1. It is a little bit expensive to use CAM for identifying those label-correlated regions. I would like to see the OOD detection performance with the randomly generated masks. For example, randomly masking 30%-70% of the image for smoothing training.
2. The proposed SMOT utilizes data augmentation for OOD detection. Therefore, the author should introduce and compare more related methods that investigate the effectiveness of data augmentation in OOD detection. I believe there have been many papers that exploring data augmentation for Calibration or OOD detection [1,2]
3. The SMOT framework is similar to the Outlier Exposure (OE) framework, the author should also compare the proposal with other Outlier exposure (OE) based methods, and discuss the advantages compared with the OE framework.

[1] RankMixup: Ranking-Based Mixup Training for Network Calibration
[2] OUT-OF-DISTRIBUTION DETECTION WITH IMPLICIT OUTLIER TRANSFORMATION

**Questions:**

1. What model is used in Table 2/3/4?

---

> ### Author Response · Authors · 2023-11-23
>
> We sincerely thank you for your constructive comments! Please find our responses below.
> >**Q1:** It is a little bit expensive to use CAM for identifying those label-correlated regions. I would like to see the OOD detection performance with the randomly generated masks. For example, randomly masking 30%-70% of the image for smoothing training.
>
> **A1:** Thanks to your kind suggestion. Following your constructive comments, we have added the experiment, with the randomly generated masks. More detailed results and disucssions can be found in **A3** to **Reviewer H2kX**.
>
> >**Q2:** The proposed SMOT utilizes data augmentation for OOD detection. Therefore, the author should introduce and compare more related methods that investigate the effectiveness of data augmentation in OOD detection. I believe there have been many papers that exploring data augmentation for Calibration or OOD detection.
>
> **A2:** Thank you for your kind suggestion.
>
> For the first paper RankMixup[1] you mentioned, it does have similarities to our work at the top level of thought. It argues that mixed samples should have lower confidence and that the higher the mixing, the lower the confidence. In our work, we think that the less complete the sample, the lower the confidence. We take inspiration from the human perspective and design our algorithm accordingly. We believe that our approach more closely aligns with how humans perceive the world However, since RankMixup is a relatively new piece of work, the authors have not published its code for the time being and we have not compared it with SMOT. As for the other mentioned work DOE[2], it uses a min-max learning scheme-searching to synthesize OOD data that leads to worst judgments and learning from such OOD data for uniform performance in OOD detection. The biggest difference between us and them is that we make a smooth excess between ID data and OOD data. We compare DOE and SMOT on CIFAR10. The result is as follows:
> |      | FPR95 | AUROC |
> |------|-------|-------|
> | DOE  | 5.15  | 98.78 |
> | SMOT | 15.42 | 97.15 |
>
> As can be seen from the experimental results, SMOT is not as good as DOE. However, this is not a fair comparison. DOE belongs to the Outlier Exposure (OE) framework and requires surrogate OOD data. while SMOT does not require any additional data. We will add a discussion of these outstanding works in the revised version.
>
> >**Q3:** The SMOT framework is similar to the Outlier Exposure (OE) framework, the author should also compare the proposal with other Outlier exposure (OE) based methods, and discuss the advantages compared with the OE framework.
>
> **A3:** We agree with your point. Our approach is indeed similar to OE. Masked samples can be viewed as OOD samples.
>
> We would like to highlight the differences between ours and OE. Specificaly, we do smoothing between ID samples and OOD samples. The more ID samples are masked, the more it is considered an OOD sample. This makes the transition between ID and OOD smoother. Also, our OOD samples are created by simply masking ID samples without the need for an external dataset. We compared our method with some OE methods with the following results:
> |       | FPR95 | AUROC |
> |-------|-------|-------|
> | OE[3]    | 12.41 | 97.85 |
> | MixOE[4] | 13.55 | 97.59 |
> | SMOT  | 15.42 | 97.15 |
>
>
> As can be seen from the experimental results, SMOT is not as good as OE and MixOE. We would like to note that it's not a fair comparison.
>
> Thanks agin for your constructive comments. We will add the results and discussions to the revised paper.
>
> [1]: RankMixup: Ranking-Based Mixup Training for Network Calibration
>
> [2]: OUT-OF-DISTRIBUTION DETECTION WITH IMPLICIT OUTLIER TRANSFORMATION
>
> [3]:Deep anomaly detection with outlier exposure
>
> [4]:Mixture Outlier Exposure: Towards Out-of-Distribution Detection in Fine-grained Environments

---

### Official Review · Reviewer_H2kX · 2023-11-01

**Soundness:** 3 good
**Presentation:** 4 excellent
**Contribution:** 3 good
**Rating:** 6
**Confidence:** 4

**Summary:**

This paper proposes a new training strategy called Smooth Training (SMOT) to improve out-of-distribution (OOD) detection performance. The key idea is to apply label smoothing to perturbed inputs rather than original inputs during training. Specifically, the authors randomly mask label-relevant regions of input images identified by class activation maps. The labels for these masked images are softened proportional to the size of the masked regions. This forces the model to output lower confidence for partial inputs, widening the gap between in- and out-of-distribution examples.

**Strengths:**

* The proposed smooth training strategy is intuitive and simple to implement, requiring only small modifications to the standard training procedure.
* Thorough theoretical analysis is provided on how the commonly used cross-entropy loss leads to overconfidence, and how smooth training can mitigate this issue.
* Comprehensive experiments on CIFAR and ImageNet-200 benchmarks demonstrate SMOT consistently improves OOD detection across different base models, scoring functions, and datasets. Improvements are also shown when fine-tuning CLIP.
* Ablation studies validate the efficacy of key components like the label smoothing function and masking threshold sampling distribution.

**Weaknesses:**

* Although smooth training enhances Out-Of-Distribution (OOD) detection, there's a minor decrease in in-distribution accuracy compared to conventional training. An in-depth exploration of this trade-off could be beneficial.
* The authors employ class activation maps to pinpoint label-relevant regions for masking, necessitating a pre-trained model. Studying other perturbation techniques that don't require a pre-trained model could expand the method's applicability.
* Further analysis could be devoted to how the results are sensitive to variations in hyperparameter settings. For instance, how essential is the use of a CAM heatmap to guide masking? What's the optimal way to establish the relationship between mask size and label smoothing hyperparameter ?

================================

Comments after Rebuttal:

Thanks for your response.

1. The paper would greatly benefit from additional experimental analysis regarding the hyper-parameter $\lambda$. There is a discernible balance to be struck between in-distribution (ID) accuracy and OOD performance, which varies with $\lambda$. The sensitivity of this trade-off to different datasets is not sufficiently addressed. I encourage the authors to explore this aspect further to enhance the utility of their approach.

2. The SMOT technique appears to depend on a pre-trained model to generate the masking (noting that the use of random noise results in subpar OOD performance), a process which adds computational complexity. The paper does not elaborate on alternative methods for 'dirtying' clean images beyond masking or the application of random noise. Exploring and discussing potential alternative techniques would provide a more comprehensive understanding of the method's applicability and limitations.

Given these considerations, I am inclined to raise my score. I am hopeful that the authors will take these comments into account and address them in the final manuscript to strengthen the paper's contribution.

**Questions:**

see weakness

---

> ### Author Response · Authors · 2023-11-23
>
> We sincerely thank you for your constructive comments! Please find our responses below.
>
> >**Q1:** Although smooth training enhances Out-Of-Distribution (OOD) detection, there's a minor decrease in in-distribution accuracy compared to conventional training. An in-depth exploration of this trade-off could be beneficial.
>
> **A2:** Thanks for your suggestion.
>
> Smooth training does lead to a minor decrease (from 95.12% to 94.54%) in classification accuracy of ID samples. Following your suggestion, we further explore the trade-off between OOD detection performance and ID classification performance on CIFAR-10. We do this by varying the values of the hyperparameters $\lambda$. The experimental results are as follows:
> |               | Texture |       | SVHN  |       | iSUN  |       | Places |       | LSUN  |       | Average |       |        |
> |---------------|---------|-------|-------|-------|-------|-------|--------|-------|-------|-------|---------|-------|--------|
> |               | FPR95   | AUROC | FPR95 | AUROC | FPR95 | AUROC | FPR95  | AUROC | FPR95 | AUROC | FPR95   | AUROC | ID ACC |
> | w/o smot      | 58.59   | 88.59 | 55.71 | 91.92 | 50.80 | 91.80 | 57.85  | 88.70 | 32.71 | 95.33 | 51.13   | 91.27 | 95.12  |
> | $\lambda = 0.01$ | 35.93   | 94.25 | 18.92 | 96.82 | 25.74 | 96.01 | 46.27  | 91.26 | 12.38 | 97.96 | 27.78   | 95.26 | 94.72  |
> | $\lambda = 0.05$ | 27.44   | 95.15 | 12.15 | 97.57 | 14.27 | 97.38 | 31.79  | 93.73 | 2.01  | 99.35 | 17.53   | 96.63 | 94.56  |
> | $\lambda = 0.1$  | 23.01   | 96.24 | 8.27  | 98.21 | 12.27 | 97.89 | 31.52  | 93.88 | 2.04  | 99.56 | 15.42   | 97.15 | 94.54  |
> | $\lambda = 0.3$  | 37.92   | 93.91 | 26.92 | 95.81 | 28.84 | 95.67 | 45.06  | 91.79 | 21.16 | 96.81 | 31.98   | 94.79 | 94.22  |
> | $\lambda = 0.5$  | 26.24   | 95.65 | 21.05 | 96.32 | 19.68 | 96.89 | 43.83  | 91.96 | 14.09 | 97.67 | 25.97   | 95.70 | 94.50  |
> | $\lambda = 1$    | 31.96   | 94.46 | 27.23 | 95.72 | 17.25 | 97.07 | 42.03  | 91.85 | 8.53  | 98.15 | 25.5    | 95.45 | 94.02  |
>
> The experimental results show that the ID ACC of models trained with SMOT is lower than that of models trained with normal cross-entropy loss, and that larger $\lambda$ usually leads to lower ID ACC, but not necessarily to higher AUROC. In practice, we need to choose an appropriate $\lambda$ so that the model maintains both high ID classification ability and good OOD detection ability.
>
>
> >**Q2:** The authors employ class activation maps to pinpoint label-relevant regions for masking, necessitating a pre-trained model. Studying other perturbation techniques that don't require a pre-trained model could expand the method's applicability.
>
> **A2:** Thank you for your kind suggestion.
>
> In addition to using masking as perturbation function, we also use adding gaussian noise to inputs as perturbation function, which does not require additional pre-trained models. We report the experimental results in Table 9 in the Appendix:
> |             | Texture |       | SVHN  |       | iSUN  |       | Places |       | LSUN  |       | Average |       |
> |-------------|---------|-------|-------|-------|-------|-------|--------|-------|-------|-------|---------|-------|
> | Method      | FPR95   | AUROC | FPR95 | AUROC | FPR95 | AUROC | FPR95  | AUROC | FPR95 | AUROC | FPR95   | AUROC |
> | MSP         | 58.59   | 88.59 | 55.71 | 91.92 | 50.80 | 91.80 | 57.85  | 88.70 | 32.71 | 95.33 | 51.13   | 91.27 |
> | SMOT(mask)  | 23.01   | 96.24 | 8.27  | 98.21 | 12.27 | 97.89 | 31.52  | 93.88 | 2.04  | 99.56 | 15.42   | 97.15 |
> | SMOT(noise) | 41.40   | 93.05 | 36.78 | 94.64 | 27.91 | 95.87 | 44.64  | 91.40 | 23.40 | 96.77 | 34.68   | 94.35 |
>
> It is shown that adding noise can also improve the OOD detection performance of the model.

---

> ### Author Response · Authors · 2023-11-23
>
> >**Q3:** Further analysis could be devoted to how the results are sensitive to variations in hyperparameter settings. For instance, how essential is the use of a CAM heatmap to guide masking? What's the optimal way to establish the relationship between mask size and label smoothing hyperparameter?
>
> **A3:** We apologize that we missed the important experiment of comparing with random masking.
>
> As suggested by you and other reviewers, we add this experiment on CIFAR10. Since the images in CIFAR10 is of 32\*32, we divide the image into 64 small 4\*4 squares, and in each loop, we sample a probability $p$ from the distribution of $Beat(\alpha, \beta)$, let each small square be masked with probability $p$, and then smooth the label. The  smoothing parameter is designed as $\epsilon(p) = (exp(p/T) - 1)/(exp(1/T)-1)$. We conducted experiments with different $\alpha, \beta$ and T. Results are as follows(The numbers in the table are the average AUROC/the average FPR95.):
> | $(\alpha, \beta)$    | $T=10$        | $T=1$         | $T=0.3$       | $T=0.1$       |
> |-----------|-------------|-------------|-------------|-------------|
> | (1,1)     | 92.91/39.02 | 93.76/35.14 | 93.62/36.78 | 92.08/44.14 |
> | (2,2)     | 91.50/37.48 | 91.27/42.00 | 90.30/41.09 | 91.47/45.07 |
> | (2/3,2/3) | 89.38/44.34 | 90.97/41.33 | 89.31/46.17 | 92.24/41.27 |
> | (5,2)     |91.79/43.34 | 90.95/44.55 | 91.59/46.87 | 91.79/43.34 |
> | (2,5)     | 90.17/37.44 | 89.00/42.54 | 90.12/40.23 | 92.28/42.02 |
> | 50,20)    | 91.23/45.54 | 93.28/36.55 | 85.33/54.66 | 91.83/42.89 |
> | (20.50)   | 89.64/35.99 | 87.47/49.29 | 86.64/45.58 | 92.56/39.57 |
>
> As can be seen from the experimental results, when CAM is not used to guide masking, it does not work well and in many cases has lower performance than using simple cross-entropy loss. This is because masking randomly does not necessarily result in a change in the label of the image. It doesn't make sense to soften the label when we mask the background
>
> As for how to establish a relationship between the masking size and label smoothing hyperparameter, this is really a major drawback of SMOT at the moment. In our paper, we simply use a temperature-scaled exponential function with a adjustable temperature. A better approach might be to get the relation between masking size and smoothing hyperparameter from priori knowledge or to learn one such relation. This is our future research direction.

---

### Meta-Review · Area_Chair_gnDi · 2023-12-10

**Metareview:**

This paper proposes a label smoothing based training framework for OOD detection, called Smooth Training (SMOT). The key idea is to apply label smoothing to perturbed inputs rather than original inputs during training. Specifically, the authors use class activation maps to identify the regions that have a strong correlation to the true label, and randomly mask out those regions. The labels for these masked images are softened proportional to the size of the masked regions. This forces the model to output lower confidence for masked inputs. Experiments on CIFAR-10/100 and ImageNet-200 show that the smooth training strategy improves the OOD performance with different score functions over certain existing methods.

While the results look promising, the idea of training with perturbed data for better calibration of certainty is not novel, moreover, to utilize it for OOD detection requires extra hyperparameters which is not ideal. The method requires a pretrained model to use CAM, as well as training all the models from scratch, as opposed to post-hot OOD detection techniques. All the incurred complexity makes the method less favorable.

The reviewers give very reasonable suggestions to improve the paper, including additional experimental analysis regarding the hyper-parameter, as well as moving away from using CAM. More careful comparisons to closely related methods like OE and RankMixup would also be beneficial.

**Justification For Why Not Higher Score:**

While the results look promising, the idea of training with perturbed data for better calibration of certainty is not novel, moreover, to utilize it for OOD detection requires extra hyperparameters which is not ideal. The method requires a pretrained model to use CAM, as well as training all the models from scratch, as opposed to post-hot OOD detection techniques. All the incurred complexity makes the method less favorable. Overall, this work's contribution is mainly engineering tricks on many levels, and does not yet meet to bar of ICLR.

**Justification For Why Not Lower Score:**

N/A

---

### Decision · Program_Chairs · 2024-01-16

Reject